# Documenting Urban Morphology: From 2D Representations to Metaverse

Alessandro Merlo * and Gaia Lavoratti

Department of Architecture, University of Florence, 50121 Florence, Italy; gaia.lavoratti@unifi.it
* Correspondence: alessandro.merlo@unifi.it; Tel.: +39-335-5423014

**Abstract:** The documentation of urban morphology is linked to the complex operation of representing the city, which over the centuries has been undertaken using different methodologies, instruments, and purposes. The "IT revolution" has expanded the possibility of overlapping and relating multiple pieces of information in connection to the urban organism on the same support and, on the other hand, has opened up new scenarios linked to the use of urban digital twins to support the analysis and urban planning. The 21st century has marked a momentous turning point compared to the recent past: the advent of artificial intelligence has in fact allowed the introduction, alongside Urban Information Systems, of 'Predictive' Systems, capable of formulating new scenarios on the basis of the elements available and pictured on 3D models. At the same time, the technical and technological acquisitions of the last century have contributed to evident experimentation on the metaverse, which, although it still exists in a de-emphasised form, is currently a whole universe under construction and expansion. Its rules are written with every passing day, in which the individual can recreate a reality similar to, or absolutely antithetical to, the one they experience on a daily basis, populating virtual cities that elude the established urban dynamics of physical structures.

**Keywords:** urban representation; urban analysis; 3D digital model; urban digital twin; information communication technology; city information modelling; artificial intelligence; metaverse

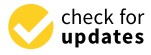



"The act of representation is thinking and building simulacra, the simulacra of what exists, has existed but no longer exists as it did at the beginning of its life, or at some other stage, and that could be restored to its primary image. It is also a simulacrum of what will, or will not, exist. The act of designing therefore also evokes what is invisible and impossible. Representing is then a substitutive act. It creates a vicarious entity located in a space where something that [...] cannot really be present at a certain point in space and time. However, a design does not exhaust its role in the creation of these supplementary images, these surrogates. [...] At the very moment of the creation of a deferred replica of an object, a second reality is in fact defined, a parallel universe in which the simulacrum is as real as the object to which it refers to" [1].

The act of depicting a city is a complex operation that involves the analytical documentation of multiple and heterogeneous factors, their subsequent reading through cultural filters and, finally, their critical representation through the adaptation of the reality to codified and shared graphic symbols. Over the centuries, as the terms of this operation have changed with different purposes, so have the tools used for analysis and their figuration, or the different *modus operandi* of those who have engaged in it.

The main goal of this paper is to underline that the information on a settlement is still 'closely linked' today, as it was in the past, to its (first analogue, then digital) representations of the reality investigated. Furthermore, we aim to highlight the possibilities inherent in the use of responsive and predictive 3D models in the management of a city through the most advanced facility management software.

## 1. Introduction

*The Role of Knowledge in the Representation of the City*

"It is interesting to note that the observations of cities experiencing incessant changes and developments, although of a particular nature, constitute the original nucleus of ever more extensive observations on urban phenomena. For the first time, it teaches us to see the city from within, in the multiple manifestations of its activity, therefore creating the first context of an urban discipline that is gradually more aware of its tasks and functions" [2].

The understanding of the city necessarily goes through the decoding of its complexity [3]. The survey, in its maximum semantic extension[1], constitutes the main critical investigation instrument capable of describing the material components in the first instance and, progressively also the immaterial ones that characterise it, in order to support any decision in a conscious manner.

The difficulty in depicting an urban organism in an unambiguous and exhaustive manner is therefore inherent to its very definition as an organised system of various parts[2] that are interconnected and dependent on the dynamics to which that society is exposed.

In this regard, it is possible to discern significant differences between the pre-industrial, the modern, and the contemporary city.

Until the First Industrial Revolution, the changes produced at different times and in different ways in the community to respond to new socio-political-economic needs only marginally concerned the built environment, succeeding in adapting to and grafting itself into a consolidated and historicised urban structure [6][3]. However, since the second half of the 19th century, economics and politics, and more recently finance, have instead produced new phenomena (often caused by induced necessities) that have contributed to a sudden expansion of the city's boundaries (the so-called urban revolution). This imposed new models that have not yet been matched by a new culture of living [4]. In turn, the contemporary city—veteran of the profound transformations that took place in the previous two centuries—is confronted with the consequences of the so-called "IT revolution" and the processes of globalisation that have accentuated its critical aspects. The gradual transfer of the exclusively physical interactions into a virtual environment increasingly pushes the individual to distance themself (or lose interest) from the urban context understood in the traditional sense of the term.

Nowadays, the growing texture of heterogeneous contents enclosed in the urban space, which is becoming dense with relations, makes it increasingly necessary to involve a multi-disciplinary team of professionals capable, each within their own sphere of competence, of conducting targeted analyses on specific aspects, which only once they are recomposed in a single cognitive framework will make it possible to provide the most complete and exhaustive description of the reality analysed[4].

Without delving into specialised studies such as those on the perception of public space conducted in the 1960s [7,8] and 1970s [9], it is nonetheless worth emphasising how the definition of an urban environment cannot lie outside the contribution made by its demographic, ecological, and climatic components. The definition of the *Space syntax*[5] analysis methodology developed in the 1980s constitutes an emblematic experience carried out with the main purpose of investigating the 'social role' of the space; it has contributed in the following decades to the formalisation of studies on the synergies between networks, places, and people [10] (Figure 1).

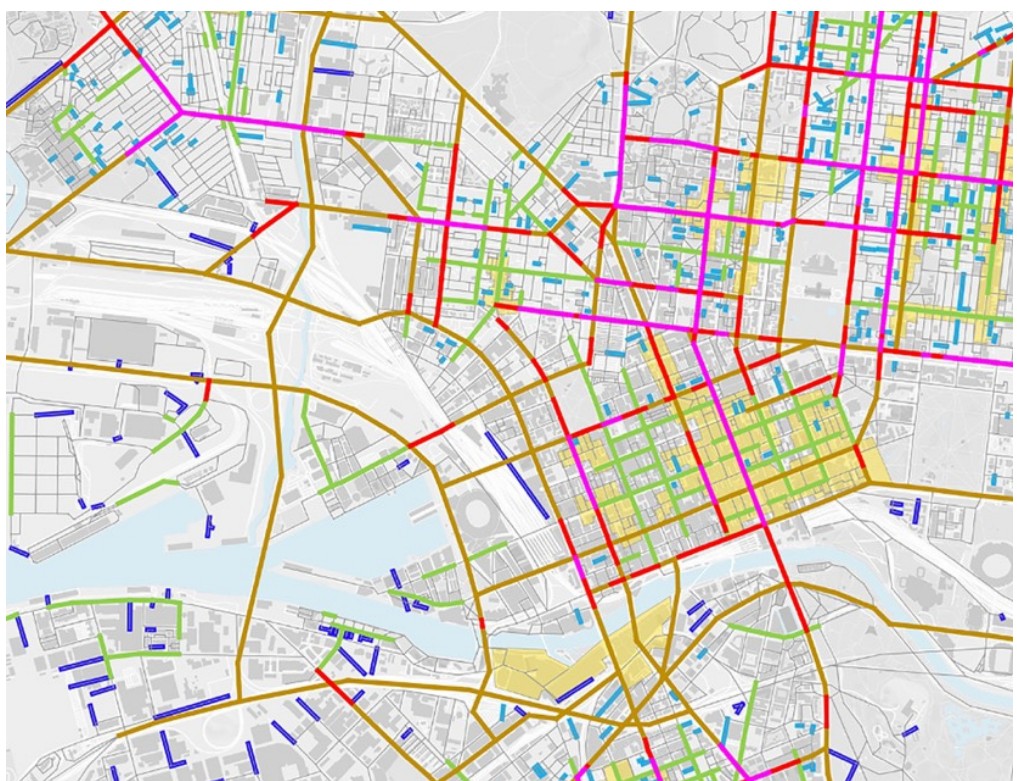

**Figure 1.** Space Syntax, Melbourne Urban Value Modelling (Project Director Nora Karastergiou, Project team Sofia Hoch). Application of the space syntax analysis methodology to the urban context of Melbourne for measuring the impact on land values of key urban design factors including spatial connectivity, land use attraction, and transport infrastructure (image taken from Space Syntax, with permission from Tim Stonor (member of Space Syntax's Board of Trustees, Board of Directors and Management Group) https://spacesyntax.com/project/melbourne-urban-value-modelling/, accessed on 12 January 2024).

At the same time, the expansion of the amount of data to be managed frequently requires the use of specialised instruments and technical supports that are capable of gathering, processing, and storing information about the urban organism. The application of a sensing and responding approach[6], aimed at giving the city a 'smart' dimension[7], has contributed, for example, to the widespread endowment of public spaces with technological devices capable of monitoring and profiling urban users[8], as well as the diffusion of the Living Lab[9].

## 2. Materials and Methods

### 2.1. The Representation of the City—2D and 3D Figurations

"Thus, the renewed art of town planning has to develop into an art yet higher, that of city design—a veritable orchestration of all the arts, and corresponding needing, even for its preliminary surveys, all the social sciences. Here, then, is the problem before us on our return to survey our modern towns, our ancient cities anew, to decipher their origins and trace their growth, to preserve their surviving memorials and to continue all that is vital in their local life, and on this historic foundation, and on a corresponding survey and constructive criticism of our actual present, go forward to plan out a better future with such individual and collective foresight as we may" [14].

It is well-known how bidimensional drawings—essentially plans, elevations, sections and floor plans, that since the second half of the 19th century have been scientifically depicted thanks to Monge's operations of projection and section[10]—make it possible to describe the morphometric characteristics of the city or its parts, and only certain phenomena/attributes that on this basis have the possibility of being graphically represented

(e.g., intended use, state of conservation, building types, etc.). Even today, upon closer inspection, the majority of the urban analyses and planning operations are represented almost exclusively on zenith views, so much so that the image of the city is sometimes confused with its plan form [15].

For decades, 2D representations have in fact played the dual role of a document that offered itself to numerous readings and interpretations (such as those relating to the transformations that have taken place over time)[11], but also of a database ahead of its time (i.e., a support on which to represent graphically (store) other data derived from the analyses conducted in different fields such as demography or sociology. This role will be taken over by 3D digital models, exponentially expanding their potentialities, as explained in this paper.

It is important to emphasise how the ways of representing a city have evolved concurrently with the progressive realisation that the essence of a heterogeneous organism such as a settlement is not given by the simple sum of the structures found in it, but by the synergies they manage to establish among themselves; that is, by the continuous interaction in time and space between the different components, both material and immaterial.

The drawing of a city, in fact, cannot end with the description of the geometry, dimensions, positions, and main characteristics of its physical elements and all the other information grasped from the morphometrical data, but should also highlight their relations, the connections, and reciprocal existing relationships. At the same time, it needs to document that the functions, destinations, vocations of use, and modes of fruition have an impact on the spatial configurations and are determined by the humans who inhabit them, travel through, and use them and by environmental conditioning.

With the advent of information technology, particularly graphic information, a new and different way of depicting the city using 3D models, or rather digital copies, and of visualising and interacting with them through devices (from simple screens to CAVEs) has been flanked and progressively imposed, and which in fact today constitutes the 'advanced' support that is replacing the traditional sheet of paper. The 3D virtual models, which for at least two decades have already constituted the means through which architects, town planners[12], and planners analyse the reality that surrounds us (morphometric documentation operations of an artefact are carried out starting from its digital twin and increasingly rarely on the real object), have more recently assumed, thanks to facility management programmes, a central role in the management of information (in some cases real big data) relating to individual buildings as well as to cities or the territory [17,18].

As it is easy to infer, while relating multiple and heterogeneous data into a single mosaic of knowledge is a possible course of action, albeit costly in terms of time and resources, representing and visualising them simultaneously on the same medium can become a technically complex operation that requires the identification and sharing of modalities established *ad hoc*. The depiction of a piece of a city by means of analogue instruments has always needed a simplification of forms linked to the scale of restitution, which, in order not to turn into a loss of significant data, has had to be accompanied by a symbolic integration of the non-drawn information.

The apparent incongruity in depicting urban 'space', by definition three-dimensional, through a series of two-dimensional drawings was overcome in the 1970s using representation codes (cf. Turin experience, which gave rise to the standard UNI 7310:1974 "Urban cartography. Conventional representation of historical aggregates prevalently distinguished by multiplane building" [19] or the Naples one in axonometry[13], Figures 2 and 3). This expedient made it possible for the knowledge levels of the city to be materialised into superimposed layers on the same planimetric base and the design indications were synthesised and graphically represented with plan markings [22,23]. This mode of representation has accompanied the drawing of the city for decades and has persisted to the present day, maintaining intact its communicative value of analytical synthesis.

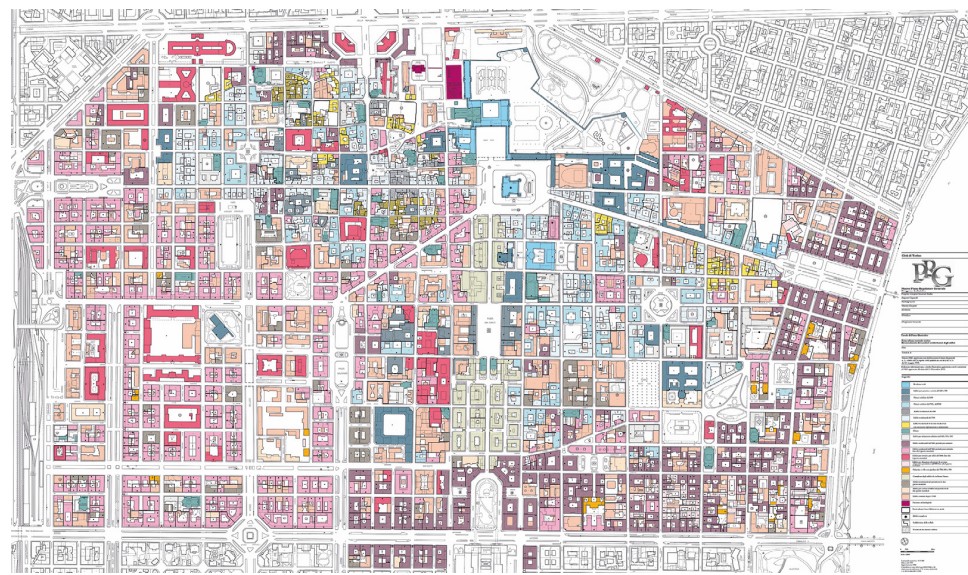

**Figure 2.** Table no. 6 "Historic central urban area. Recognition of the architectural characteristics of buildings" of the new General Town Plan with application of the graphic regulations implemented by UNI 7310/74, original scale of representation 1:2000 (Città di Torino, Nuovo Piano Regolatore Generale, 1995, at http://geoportale.comune.torino.it/web/governo-del-territorio/piano-regolatore-generale, accessed on 12 January 2024).

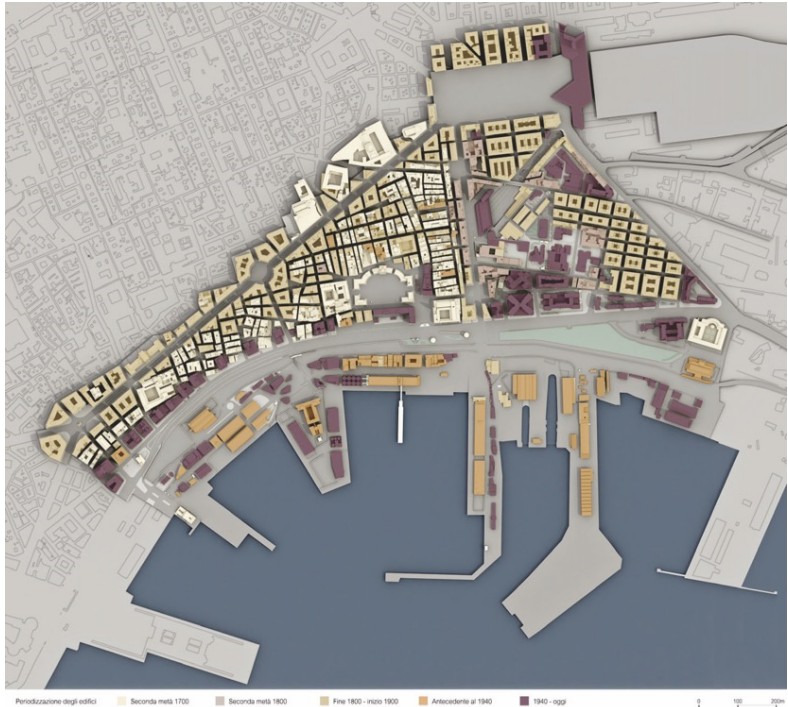

**Figure 3.** Views of the three-dimensional model of Naples developed by the working group coordinated by Prof. Riccardo Florio in 2010, in continuity with the experience of "Naples in Axonometry" coordinated by prof. Adriana Baculo and published in 1992. The working group coordinated by Prof. Riccardo Florio is composed of: Teresa Della Corte, Raffaele Catuogno, Carmen Frajese D'Amato, Simona Cuomo, Alessia Mazzei, Morena Beatrice Mennella, Sergio Migliaccio and Giovanna De Fazio. Volumetric plan of the urban area of the eastern coastal strip, with indication of the main historical periods. Complementary axonometric views from the south and the north of the investigation area as a whole (image taken from [20] with permission from Riccardo Florio).

The undeniable ability of axonometric and perspective views to convey information related to the spatial perception of the city has been recovered using 3D models [23]. The transition from analogue to digital, both in the acquisition and restitution phases, makes it possible to create, in a short time and with millimetric precision, a three-dimensional model of the existing artefact (digital twin)[14] on a 1:1 scale on which, a posteriori, it is possible to link databases containing information on different areas. These make it possible to conduct a potentially unlimited series of analyses and, subsequently, to pour over as many thematic readings of the built environment, set up design simulations, and monitor the consequences of the latter.

Information communication technology makes it possible, thanks to CIM platforms (*city information modelling*)[15] to operate with transcalarity, making all of the data connected to a phenomenon (or an object) from the general to the particular, and vice versa, binding them to 3D reality based (urban digital twin, UDT)[16] models, created at times with a different level of detail (LoD).

In other words, "Knowledge of the city, the territory and the environment in the ICT era is a field in which major changes have been taking place for several years now. The digital world of communication is favouring the development of a new cognitive approach based on the realisation of digital models whose content, increasingly dense and articulated, derives from the integration of consolidated knowledge with information acquired with sensing technologies" [26–28].

To avoid any misunderstanding, it should be noted that, as a mere representation of reality, the digital twin constitutes an 'uncritical'[17] virtual reproduction, a formally and metrically 'exact'[18] copy of what exists, of which nothing has been revealed yet. In this sense, it represents a neutral field on which the multidisciplinary team can investigate the reality remotely.

The possibility of representing heterogeneous and previously hardly comparable data on the same digital model UDT has made CIM (city information modelling) a valuable tool for urban analysis. The interaction between the information underlying the functioning of CIMs, in fact, is what until a few years ago was closest to the relationship between the physical and human components within the urban environment [29–31].

In such complex databases, alongside the categories and sub-categories dedicated to maps and the dimensional-spatial definition of the built environment, are found more and more information relating to the economy, demographics, population density, health, and culture, etc. The city is in fact, by definition, a demographic space, built by humans for humans, modelled for their needs and for those of the context in which it belongs and, as such, it is constantly changing. The community modifies the urban space, not only its superficial appearance, but in the long-term, its identity features. For this reason, comprehensive documentation must integrate the specialist analyses conducted on the built environment and illustrated above with a careful reading of the parameters relating to human action, at least of those that are objectively quantifiable.

Still being an experimental form, CIMs now make it possible, thanks to innovative technologies, (Internet of Things, high performance computing, data science, advanced optimization, etc.) to implement information and, consequently, to modify UDTs (in this case responsive models). These models, therefore, constitute tools capable of "not only helping people better understand the city but also assist in decision-making at all stages of the city life cycle. Therefore, the CIM visualizations integrating VR and augmented reality (AR) have drawn the main research concerns" [25]. The responsive models contribute, as a matter of fact, to the formation of a cognitive framework that, although never exhaustive but constantly implementable, can be visualised on the same support that can be navigated, questioned, and thematised [32] (Figure 4).

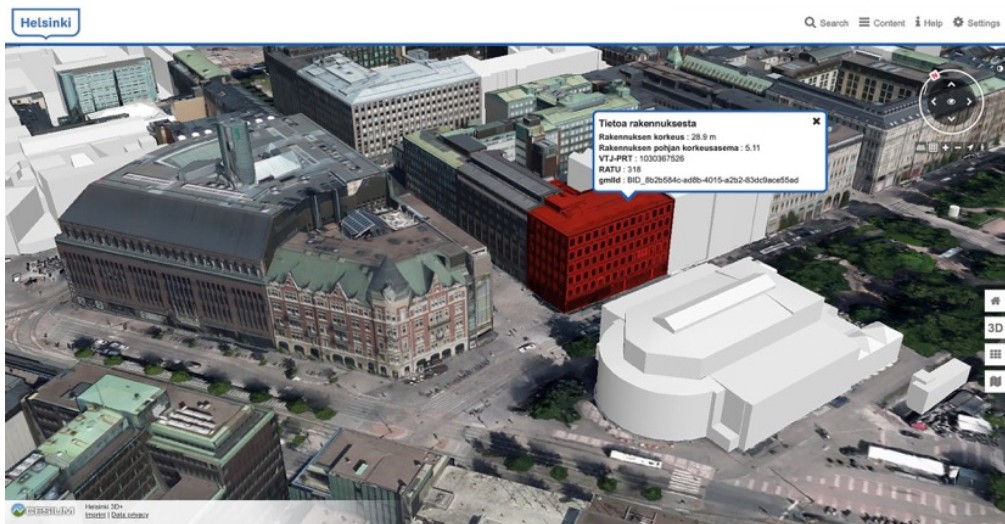

**Figure 4.** Digital twin project in Helsinki city. The city of Helsinki has been experimenting with new digital modelling solutions and applying the cityGML standard in its 3D city modelling work. The 3D model of Helsinki includes two parts: a numerical model and a semantic model based on CityGML. The numerical model, a visually high-quality reality mesh model, was created from point clouds generated using laser scanning and digital photogrammetry. The point cloud data were processed, optimised, and visualised to create a huge 3D model of the entire city transmitted to web browsers and device furniture, so that the city could be visualised on any mobile platform that supports AR environments or VR38. The semantic model includes the various analyses (static data and dynamic data) were carried out on the city. The digital twin project in Helsinki city is open to the public (https://kartta.hel.fi/3d/#/); it therefore constitutes an important tool for the population who, thanks to CIM, can use urban data and contribute to collecting it, thus keeping the model constantly updated, thanks also to the application of IoT. As a result of this testing project, it was noticed that compared to static CAD models, the cityGML standard takes 3D urban design and planning to a higher level. Further experimentation of the CIM model was carried out for NTU EcoCampus, where 200 buildings covering a surface area of 1.1 million m$^2$ were equipped with sensors capable of acquiring data and optimising the energy management of the campus in real-time [24].

The application of a sensing and responding approach to a digital model thus conceived makes it a true hypermodel that can be integrated within the ecosystem, acting as a hinge between the virtual and real space.

Such a complex and delicate operation presupposes a paramount implication: the progressive and inevitable transfer of control of the inputs as well as the evaluation of the outputs from a human component to an automated component, whose actions must constantly be verified and validated.

### 2.2. Towards New Scenarios

With the arrival of artificial intelligence (AI), it is, in fact, possible to have a clear distinction between information systems based on static models and those based on dynamic, responsive, and predictive models. The former are limited to data visualisation, while the latter, through the use of city simulation platforms, the direct user interaction, and the use of sensors, allow for new scenarios to be formulated on the basis of the available elements and visualised on 3D models.

By analysing information about the status quo and past events, and thanks to the support of artificial intelligence and machine learning, predictive analysis aids in the understanding of what is likely to happen in the future. These devices, which are already widely used in the medical field, business processes, and e-commerce to name a few, are more recently also affecting the city. Some examples are predictive models applied to

logistics and transport, real estate management, water services, energy supply, cleaning, and waste collection [33].

The extent to which these systems are likely to influence the formal as well as the functional design of the contemporary city, and even more so the future city, will depend on our ability to creatively use the tools that tend to be deterministic, since they are based on deductive, stochastic, and probabilistic logic [34] (Figure 5).

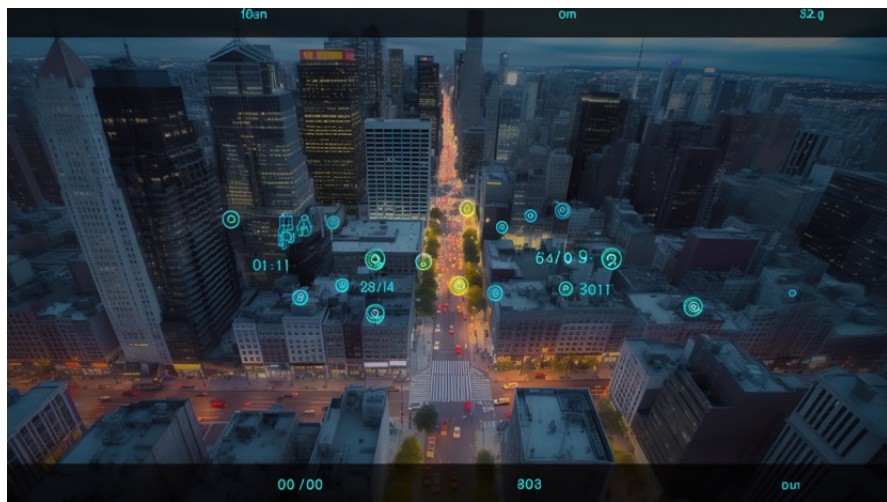

**Figure 5.** Predictive system for the city (image taken from Freepick, by kenshinstock, free license).

However, while the realisation of twin digital cities allows for the better management of real ones, we are increasingly witnessing the progressive distancing of citizens from tangible places of aggregation in favour of their virtual abstractions, developed following the massive use of social media platforms.

It is irrefutable, indeed, that virtual reality (VR) in all of its possible declinations (AR, MR, IR, etc.) is increasingly pervading our existence, so much so as to pose itself, in some cases, as an effective alternative to reality [26] (Figure 6).

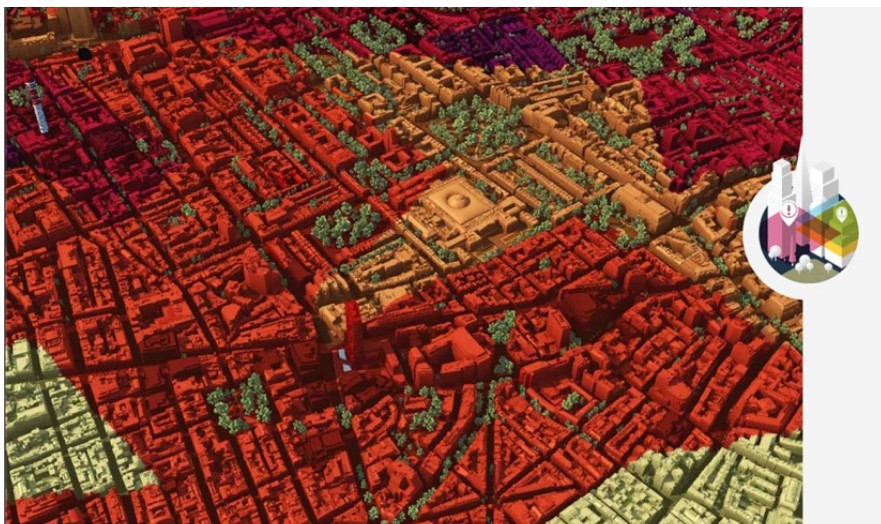

**Figure 6.** VU.CITY London. VU.CITY London is the interactive model created for the English capital as part of the London Plan of the Greater London Authority (GLA). This accurate digital twin constitutes a fundamental support in city planning and in directing policies relating to the volume, density, and height of new interventions. VU.CITY London covers the entire 1619 km$^2$ territory with an accuracy of 15 cm, with a combination of features and data (https://www.vu.city/cities/london).

The consideration to be made concerns, therefore, the apparent paradox that the digital transition is highlighting: the possibility of obtaining complex databases and interactive models that are increasingly closer to the reality of 'urban' organisms that, at least in their social component, tend to gradually move away from the physical space.

Leaving aside the pathological conditions typical of the most common forms of social anxiety, from FOMO (fear of missing out)[19] to Hikikomori[20], just to name a few, while the assessment of their dramatic impact on the community was beyond the scope of this paper, it is inevitable to emphasise how the modes of interpersonal sharing have changed and, at the same time, are transforming the intrinsic meaning of 'being together', giving it a dystopian and alienating meaning. If, on the one hand, the typical socio-cultural fragmentation in the contemporary city, far from being integration, contributes to the stripping of public spaces of their inherent value as places of cohesion, on the other hand, the 'virtual transition' of interpersonal relationships amplifies the existing gap between physical spaces and spaces to foster interaction.

The nodal element par excellence in the complex structure of the urban settlement, the square, historically represents the place where most of the functions linked to collective life, but in recent decades, has been stripped of its aggregative role. From vital point to marginal sphere, in a collective logic that privileges new kinds of squares, the social media ones, in which to recreate or virtually strengthen social relations complicated by heterogeneous and articulated individual dynamics. Interpersonal interactions are increasingly moving in the direction of a largely unexplored metaverse to be constructed.

Although the concept of the 'metaverse'[21] was introduced back when no one would have imagined, more than thirty years ago[22], it is only following the technical and technological achievements of the computer revolution that the debate on its actual 'reproducibility' has been given a tangible meaning, shifting the focus of the issue from the field of science fiction to that of science. If we accept as the definition of the term: "a vast digital space focused on social connection, in a hypothetical synthetic environment linked to the physical world" [37], it becomes clear how the virtual universe described is conceived in close connection to the commonly experienced material reality, of which the metaverse constitutes an extension and/or a reproduction (Figure 7).

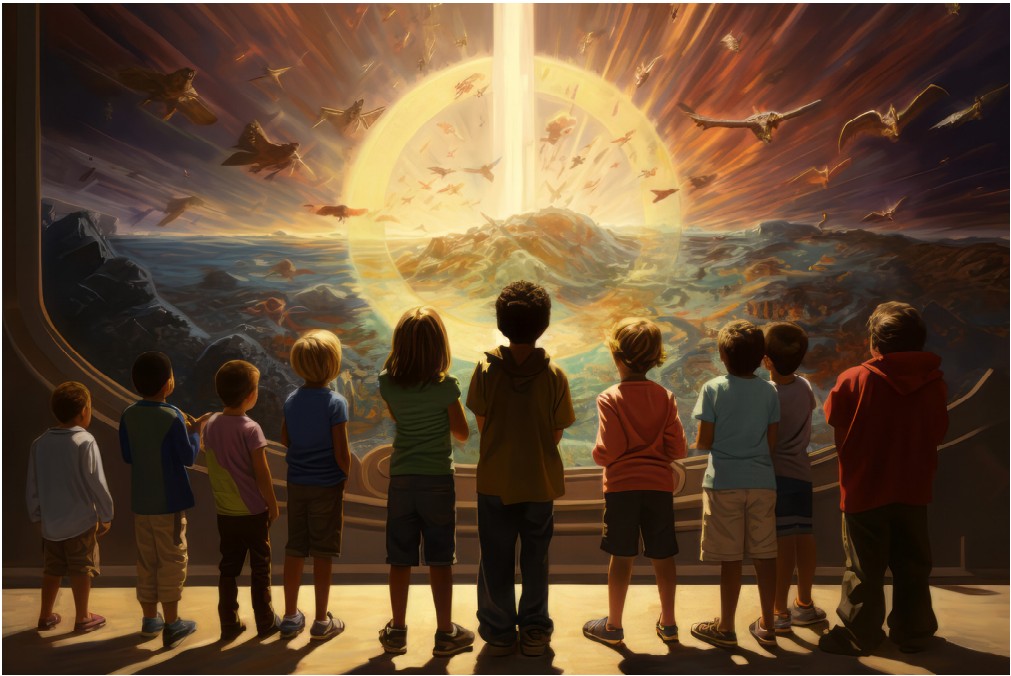

**Figure 7.** Metaverse and places of aggregation (image taken Freepick, free license).

This hypothesis can take on two distinct interpretations: the first evidently focuses on the concept of the digital replicability of tangible space, while the second brings into play AI, which is capable of creating specially made virtual environments starting from the information inferred from the profiling of users or those who will use them in the future.

In the first case, the focus is on the digital twins and their possible implications and uses, for purposes that have so far only been partially experimented[23]. From this point of view, the reality-based digital model becomes the plausible scenario within which the actions take place and the social relations established are entrusted to avatars[24] (Figure 8).

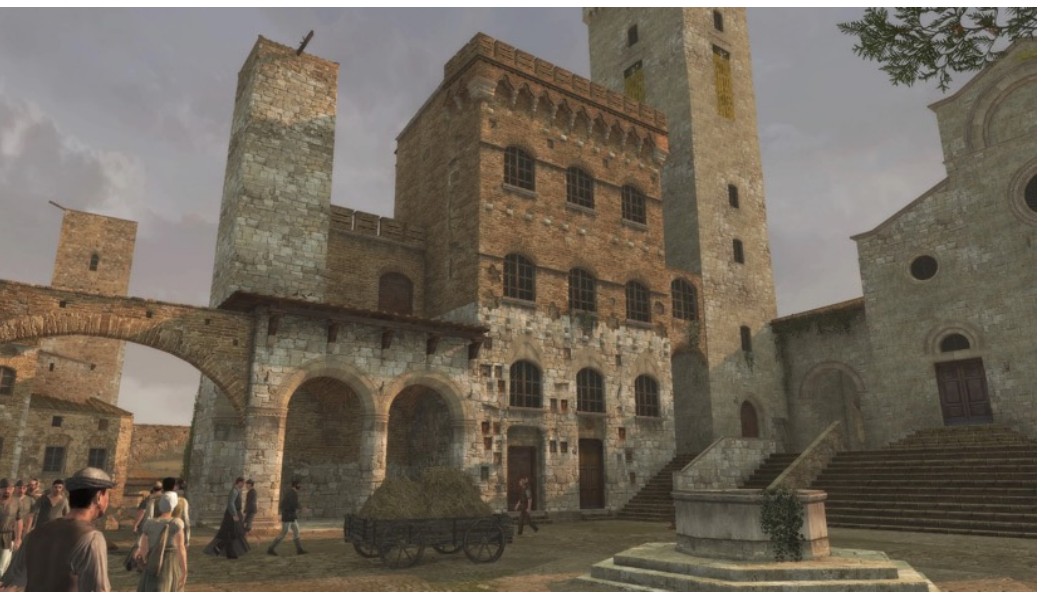

**Figure 8.** Existing cities and the metaverse. Detail of the Piazza del Duomo in San Gimignano, Siena (image taken from Ubisoft, Assassin's Creed II). The second chapter of the series is set during the Italian Renaissance. Like all chapters of the series, the creation of the settings is characterised by a very detailed graphic work that aims to represent the places in a plausible manner. The Assassin's Creed II development team personally visited locations and studied in archives, libraries, and museums. They used maps, paintings, and books from that time to reconstruct the atmosphere of the time, also involving historians and art historians.

The second meaning focuses on the semantic evolution of the term 'metaverse' over the years, which has seen a gradual departure from a strict relationship with real and recognisable [39] geographical settings towards spaces designed and customisable by the user [40] (Figure 9).

The metaverse, in fact, although it still exists in a de-emphasised form[25], is currently a universe under construction and expansion, whose rules are written every passing day, in which the individual can recreate a reality similar to or absolutely antithetical to the one they experience on a daily basis, populating virtual cities that elude the established urban dynamics of physical structures.

For cultural professionals involved in the planning and design of human settlements, understanding these organisms will in the near future become a necessary, though not sufficient, condition to ensure that the urban environment continues to meet the (real, though often artificially induced) needs of those who inhabit it.

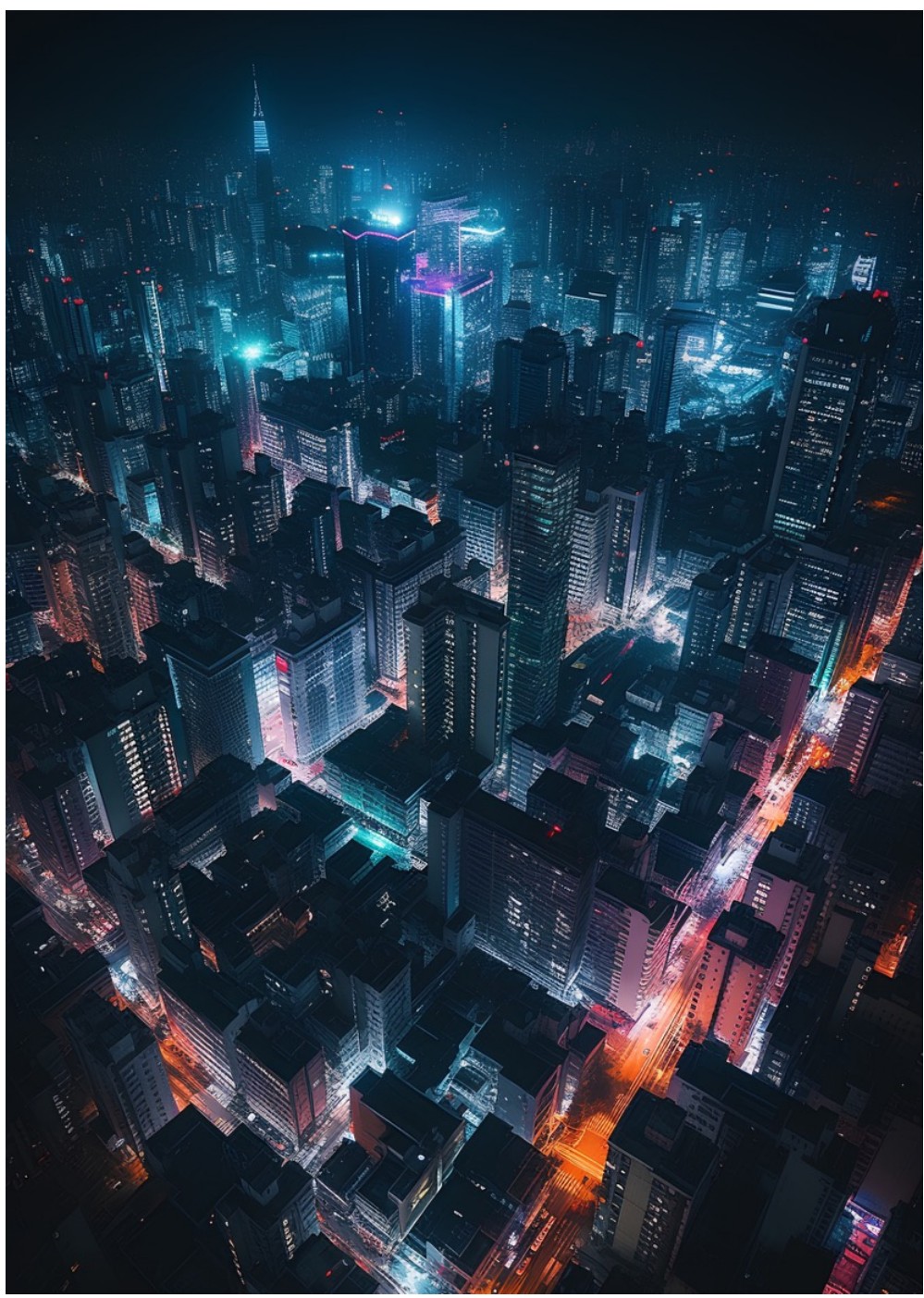

**Figure 9.** Futuristic city. Image generated by AI (image taken from 123RF Free Images, free license).

### 3. Results

The ongoing transition from static to dynamic digital twins applied to an urban organism is only possible thanks to the technological and IT progress that has affected specific sectors (from sensor technology to domotics). This has called into play multiple and varied skills that are contributing to the transformation of the city in an increasingly 'smart' direction. However, if technological and computerised systems are ever more proving themselves capable of satisfying and solving contingent issues, a conclusive consideration on the purposes and ways in which these solutions will be adopted cannot be overlooked.

Firstly, the widespread application of automatisms, both in the data acquisition and reading phase, brings to light an initial criticality linked to the control and selection of

the information. In particular, the application of a sensing and responding approach, in the absence of a precise interpretation and reading of the results, runs the risk of automatically pouring partial or false information into the city information model, which is difficult to verify a posteriori. The only objectivity of the application of pre-established parameters, in fact, cannot be sufficient to guarantee the correct detection of the data and their comprehension.

Secondly, resorting to the mechanical simulation of a thought process that, by definition, constitutes a human prerogative can be, especially in the field of urban design, a contradiction. The creative thinking behind the project cannot be replicated and simulated through the application of mathematical algorithms as it cannot be summarised through the synergy of acquired notions and problem-solving strategies. At this point, the still unresolved doubt concerns the actual opportunity to make this profound ethical and social change required in the name of technology.

**Author Contributions:** Conceptualisation, A.M. and G.L.; Methodology, A.M. and G.L.; Formal analysis, A.M. and G.L.; Investigation, A.M. and G.L.; Data curation, A.M. and G.L.; Writing—original draft preparation, A.M. and G.L.; Writing—review and editing, A.M. and G.L. In particular, paragraphs: 1. Introduction and 2.1 The representation of the city: 2D and 3D figurations, are to be attributed to Alessandro Merlo; paragraph: 2.2 Towards new scenarios and 3. Results, are to be attributed to G.L. All authors have read and agreed to the published version of the manuscript. **Funding:** This research received no external funding.

**Data Availability Statement:** The raw data supporting the conclusions of this article will be made available by the authors on request.

**Conflicts of Interest:** The authors declare no conflicts of interest.

## Notes

1. "Over time, the meaning of the word survey has extended to embrace all types of investigations aimed at gaining the 'most complete' knowledge possible of the organisms being analysed (from the description of their historical background to their state of conservation), thus altering the original meaning of the term, that is now interpreted as the critical result of transversal investigations. [...] When reference is made to the sphere of knowledge nowadays, it is therefore preferred to use the term 'documentation', subsequently declining this word with expressions that delineate the disciplinary field within which this operation takes place: morphometric and chromatic characteristics, pathologies of degradation and instability, structural, compositional, and perceptive aspects, etc." [4].

2. Guido Canella defines the city as a "*living organism*, in the innate breath emanated by the genetic, physiological endowment on the territory; and therefore [...], beyond all appearances, it possesses a hidden structure—*structure and superstructure*, as he used to say—a real skeleton that is resistant over a period of time to support, as long as it can, the cartilages and the connective tissues; and it is only from this skeleton that it can be adjusted to maintain its coherence to its role in the development and contraction, transformation and preservation of its structure" (cf. [5]).

3. Historically, the city has evolved and transformed consistently with the socio-political and environmental events to which it has been subjected. However, it is known that the social dynamics and, more generally, the totality of phenomena linked to the anthropic transformations of space occurred with different timing and speed compared to the real transformation capacity of the built environment.

4. In this context, the urban survey, in the strict sense of the term, is an operational tool capable of recognising and delineating, following measurement operations, the morphological and dimensional characteristics of a built environment as well as describing, using photogrammetry procedures (first analogue, now digital), its chromatic characteristics.

5. Bill Hillier defines Space Syntax as "a set of techniques for the representation, quantification, and interpretation of spatial configuration in buildings and settlements. Configuration is defined in general and, at least, the relation between two spaces considering a third, and, at most, as the relations among spaces in a complex taking into account all other spaces in the complex. Spatial configuration is thus a more complex idea than spatial relation, which need invoke no more than a pair of related spaces" (cf. [10]).

6. Sensing and responding refers to a "process of automated attribution of pre-established reactions to detected conditions. The implementation of an automated real-time adaptive response process requires the availability of two elements: a system capable of monitoring the building in a capillary and continuous way (smart applications for monitoring) and a response model to the detected conditions, capable to process high-volume, high-speed, and high-variety information assets" (cf. [11]).

7. A Smart City is "a city in which all resources are accessible through an efficient telematic network infrastructure and where information services are available through which the citizen and the administration can communicate" [12].

8   Profiling analysis consists of obtaining (through data collection and processing) knowledge about a group of individuals or an individual including habits, preferences, and information obtained from 'digital' interactions with reference to political, musical, and social issues that also include the network of friendships and acquaintances, and much more (cf. [13]).

9   The European Network of Living Labs today constitutes a reality capable of relating and coordinating the numerous ongoing projects, promoting the activities of these infrastructures aimed at experimenting with innovative technologies, even in the urban environment through the direct interaction of citizens.

10  From ancient maps to the earliest cadastral systems up to the most modern use of technical cartography, the use of a system of representation that allows for a 'top–down' control of the city has been considered the most effective means for the description of urban consistency and, consequently, for the depiction of the information surveyed. In fact, plans and layouts allow a complete verification of the dimensions of length and width, corresponding to the directions in which most human movements take place, and, if drawn up using appropriate graphic devices, are capable of synthesising complex and heterogeneous information.

11  The Italian morpho-typological school, attributable to the thought of Saverio Muratori (Modena, 1910–Rome, 1973) and that of his students, is emblematic in this sense.

12  Regarding the role that such digital copies can play in participatory urban planning processes, see [16].

13  This semiotic synthesis of reality is clear, for example, in the emblematic experience carried out by Adriana Baculo for the representation of Naples in axonometry, which for the drafting of the 63 plates, on a scale of 1:2000 (originally drawn at a scale of 1:1000) required the prior and simultaneous drafting of abacuses of the elements and markings capable of explaining, for each graphic element represented, the effective informative content (cf. [20,21]). In this logic, the image obtained of Naples does not aim to achieve a formal and dimensional fidelity of the built, but the built itself constitutes a resembling critical interpretation dense with meaning.

14  In this context, a digital twin is understood in the more traditional meaning of the disciplinary sector of drawing and survey (i.e., as a static digital copy of reality (reality based model obtained through architectural and urban digital survey operations with active and passive sensors)), delegating all the implications linked to the concept of a dynamic digital twin to the subsequent creation of an integrated model capable of accommodating the results of further heterogeneous investigations carried out on the same object (hypermodel) [24]).

15  The arrival of CIMs took place due to the possibility of linking the geospatial information databases, managed through geographic information systems (GIS), with those relating to buildings governed through BIM (building information modelling) for the newly built architectures and HBIM (heritage building information modelling) for the existing ones [25].

16  Similarly to the more generic definition of digital twin (see note 14), urban digital twin (UDT) here means the mere reality-based digital model, which only following an appropriate information enhancement can be considered a city information model.

17  Although, even the processes of the acquisition and restitution of the reality aimed at the realisation of a digital twin are in any case affected by a "critical" component linked to the operator's decisions before and after (on the operator's critical contribution in the transition from analogue to digital surveying see [4]), the term here is intended to emphasise the absence of the specific interpretative readings of the object that, on the contrary, are operated directly on its virtual model.

18  Its "accuracy" evidently depends on the margin of error of the instruments used during the acquisition and return phase as well as on the operator's ability to manage the process.

19  FOMO is a social phenomenon related to the digitisation of everyday life that corresponds to the fear of missing out, or not participating in, an enjoyable and rewarding experience involving acquaintances or friends. The attempt to satisfy the need for sociability can lead people suffering from FOMO to an abuse of social media platforms on a compensatory level.

20  Identified in the 1990s by the Japanese psychiatrist Tamaki Saito, Hikikomori (social withdrawal) is a self-inflicted social isolation that leads the individual to live segregated within a living space, renouncing any kind of human interaction. This form of social discomfort, induced by the discrepancy between imagined and perceived reality, although prevalent in Japan, is also becoming worryingly widespread in the United States and Europe (cf. [35]).

21  Created between the prefix of Greek origin 'meta', which conveys a meaning of transcendence, and the term 'universe'.

22  The first use of the term 'metaverse' is unequivocally recognised in the science fiction novel *Snow Crash* (1992) by Neal Stephenson. Here, the author describes a virtual world parallel to the real one, in which interpersonal relationships are not limited by the same shared social rules (cf. [36]).

23  The experience of Virtual Cities realised using the Virtual Reality Modelling Language (VRML) started at the end of the 1990s and, even more, the decades of experimentation in the field of edu-entertainment carried out on reality-based digital models, for instance, have allowed the video-ludic industry to develop an important production strand aimed at experiential simulation, in its own way, a precursor of the space–time interactions made possible within the metaverse.

24  An avatar is a customisable alter ego that represents the user and reproduces their actions and movements in real time (cf. [38]).

25  The still limited number of users and the lack of synergy between different platforms still prevent the metaverse from fully realising the potential for which it was conceived.

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
