# Peer review of "Documenting Urban Morphology: From 2D Representations to Metaverse"

_land, doi:10.3390/land13020136_

Round 1
Reviewer 1 Report
Comments and Suggestions for Authors
This manuscript explores technological advancements in analysing and visualising urban morphology. The manuscript starts with the two dimensional representation of cities and continues with the technological advancements in representation of urban areas through three-dimensional platforms. Several examples of urban analytics, such as space syntax, are also provided to delineate the chronological advancements in urban modeling and simulation domains. This work can be interesting, and through a systematic and chronological order the progress of the technological application in urban analytics could be shown. However, this paper fails to derive the results from a rigorous and systematic literature review. Also, the chronological or even technological orders are not accurate. For example, the role of GIS in representing and analysing urban morphology was not mentioned until the "results" section on page 9 of 15. Furthermore, the demonstrated example for an urban digital twin (3D model of Brisbane, Figure 5) is questionable. This is because there is no clear definition of urban digital twin in this manuscript. I could not see anything related to the UDT, but a traditional 3D city model.
The paper's structure should be improved substantially. The section on "materials and methods" does not clearly explain how the literature is reviewed and what the logic behind the examples and arguments is. The last section is "results" with one subsection (3.1) and the paper ends without any discussion and conclusion explaining the theoretical and practical implications of the findings.
Given these issues, this manuscript needs to be substantially revised. Specifically, the manuscript should be improved by a systematic literature review and scientific justification of arguments.
Author Response
We thank you for your careful review and would like to point out the following:
1. “This manuscript explores technological advancements in analysing and visualising urban morphology. The manuscript starts with the two dimensional representation of cities and continues with the technological advancements in representation of urban areas through three-dimensional platforms. Several examples of urban analytics, such as space syntax, are also provided to delineate the chronological advancements in urban modeling and simulation domains. This work can be interesting, and through a systematic and chronological order the progress of the technological application in urban analytics could be shown. However, this paper fails to derive the results from a rigorous and systematic literature review. Also, the chronological or even technological orders are not accurate. For example, the role of GIS in representing and analysing urban morphology was not mentioned until the "results" section on page 9 of 15.”
The structure of the contribution has been substantially modified to accommodate the indications provided by the referee.
The objective of this contribution is not to make a historical-critical examination of the technologies and tools for documentation at the service of the urban project, but rather to highlight the fact that, today as in the past, the information relating to a settlement are in fact 'anchored' to the representations (first analogue and subsequently digital) of the reality investigated, highlighting, finally, the possibilities inherent in the use of responsive and predictive 3D models in the management of the city through the most advanced facility management software .
2. “Furthermore, the demonstrated example for an urban digital twin (3D model of Brisbane, Figure 5) is questionable. This is because there is no clear definition of urban digital twin in this manuscript. I could not see anything related to the UDT, but a traditional 3D city model.”
The digital twin example inserted has been replaced with a CIM experience deemed more relevant.
However, a definition of DT and UDT has been included and the meaning with which the term is used in the paper has been specified.
3. “The paper's structure should be improved substantially. The section on "materials and methods" does not clearly explain how the literature is reviewed and what the logic behind the examples and arguments is.”
The structure of the contribution has been substantially modified to accommodate the indications provided by the referee.
4. “The last section is "results" with one subsection (3.1) and the paper ends without any discussion and conclusion explaining the theoretical and practical implications of the findings. Given these issues, this manuscript needs to be substantially revised. Specifically, the manuscript should be improved by a systematic literature review and scientific justification of arguments.”
Paragraph 3. Results has been modified, which summarizes the theoretical and practical implications of the results.
Reviewer 2 Report
Comments and Suggestions for Authors
The paper presentes all the aspects of documenting urban morphology, especially the evolution of technologies and approaches , also in terms of contemporary problems of urban science, including using AI and UDT to prepare the new solutions. Generally, I have no major comments to the content, paper is rather based on qualitative method, a heuristic one.
Please consider embedding the "living lab" concept in the content (just a mention of it and what is causes for the issues you mention in the paper).
I have only some minor comments to make the paper clean in a view of some usual "checkpoints" of the scientific paper:
Please indicate the goal of the paper since it is not included in the introduction.
Please provide a short remainder as well - how the goal was achieved in the paper (in what sequence of content).
Please provide the main conclusion., limitations and future research directions including research gaps in the last section of the paper (can be as a separate one).
I have no additional comments.
Author Response
We thank you for your careful review and would like to point out the following:
1. “The paper presentes all the aspects of documenting urban morphology, especially the evolution of technologies and approaches , also in terms of contemporary problems of urban science, including using AI and UDT to prepare the new solutions. Generally, I have no major comments to the content, paper is rather based on qualitative method, a heuristic one.
Please consider embedding the "living lab" concept in the content (just a mention of it and what is causes for the issues you mention in the paper).”
As suggested, a mention of "living labs" has been included.
2. “I have only some minor comments to make the paper clean in a view of some usual "checkpoints" of the scientific paper:
Please indicate the goal of the paper since it is not included in the introduction.”
The main goal of the paper was specified in the introduction.
3. “Please provide a short remainder as well - how the goal was achieved in the paper (in what sequence of content).”
The structure of the contribution has been substantially modified to accommodate the indications provided by the referee.
4. “Please provide the main conclusion., limitations and future research directions including research gaps in the last section of the paper (can be as a separate one).”
Paragraph 3. Results has been modified, which summarizes the theoretical and practical implications of the results.
Reviewer 3 Report
Comments and Suggestions for Authors
Dear Authors,
First and foremost, I would like to express my appreciation for your work on this intriguing retrospective analysis of the methods used in depicting urban spaces. Your article provides valuable insights into this subject. However, I have a few comments that might further enrich your work.
In your study, you utilize the concept of a digital twin. It is important to note that the definition of a digital twin is not limited to visual representation but also includes dynamic relationships between objects. In the context of the depictions you are discussing, the term does not seem to be fully applicable. Perhaps a more detailed explanation or redefinition of this term in the context of your research would be beneficial.
I commend the extensive review work you have undertaken. However, what seems to be missing is a clearly defined research objective, thesis, or hypothesis. Your work might serve as a chapter in a broader study, but as a standalone research piece, it appears somewhat incomplete. Consider adding research on how users perceive and understand space depending on the modeling method.
The statement in line 248 ("The technology allows for the visualization and analysis of development in relation to the existing urban environment") is partially true. The technology does enable visualization, but the relationship to the existing urban environment is not fully reflected at this point (this concerns the relationships between objects, elements, and users/residents of urban spaces).
As it stands, the article seems more akin to a piece of popular science rather than a research paper. Incorporating more detailed research analyses could strengthen the scholarly aspect of the work.
In conclusion, while your work is impressive and provides valuable insights, I believe that without further supplements, the article cannot be considered a standalone research piece. I hope my comments will be helpful in the further development of your article.
Author Response
We thank you for your careful review and would like to point out the following:
1. “First and foremost, I would like to express my appreciation for your work on this intriguing retrospective analysis of the methods used in depicting urban spaces. Your article provides valuable insights into this subject. However, I have a few comments that might further enrich your work. In your study, you utilize the concept of a digital twin. It is important to note that the definition of a digital twin is not limited to visual representation but also includes dynamic relationships between objects. In the context of the depictions you are discussing, the term does not seem to be fully applicable. Perhaps a more detailed explanation or redefinition of this term in the context of your research would be beneficial.”
A definition of digital twin has been included, with a specific note on the meaning of this term in the context of the research presented.
2. “I commend the extensive review work you have undertaken. However, what seems to be missing is a clearly defined research objective, thesis, or hypothesis. Your work might serve as a chapter in a broader study, but as a standalone research piece, it appears somewhat incomplete. Consider adding research on how users perceive and understand space depending on the modeling method.”
The main goal of the paper was specified in the introduction.
3. “The statement in line 248 ("The technology allows for the visualization and analysis of development in relation to the existing urban environment") is partially true. The technology does enable visualization, but the relationship to the existing urban environment is not fully reflected at this point (this concerns the relationships between objects, elements, and users/residents of urban spaces).”
The statement on line 248 has been deleted, as the fig. has been changed. 5 and, consequently, its caption.
4. “As it stands, the article seems more akin to a piece of popular science rather than a research paper. Incorporating more detailed research analyses could strengthen the scholarly aspect of the work. In conclusion, while your work is impressive and provides valuable insights, I believe that without further supplements, the article cannot be considered a standalone research piece. I hope my comments will be helpful in the further development of your article.”
The structure of the contribution has been substantially modified to accommodate the indications provided by the referee.
Reviewer 4 Report
Comments and Suggestions for Authors
The topic covered in the paper is interesting, methodologically correct and sufficiently original. The premises are clear and adequately explored in the text. The state of the art is well defined from both a historical and a technological point of view. The bibliography is extensive and well structured. Minor clarifications: Note 2: Some inaccuracies will need to be fixed. Eliminate the repetition of "gradually" in lines 94 and 96. Note 5: compared to what is said in the text, this definition seems too simplistic. Lines 128-136: It's a citation. Quote the original text and not the translation of a translation. You can find it on page 205 of the original text (https://archive.org/details/citiesinevolutio00gedduoft/page/204/mode/2up?ref=ol&view=theater&q=art+urban+planning). Note 12: Replace "vefore" with "before".
Comments on the Quality of English LanguageSome sentences are a little too long and complex, reflecting a typically Italian syntactic construction. I would suggest simplifying them. Likewise, some terms do not seem perfectly adequate. A quick revision of the text is recommended to correct such inaccuracies.
Author Response
We thank you for your careful review and would like to point out the following:
1. “The topic covered in the paper is interesting, methodologically correct and sufficiently original. The premises are clear and adequately explored in the text. The state of the art is well defined from both a historical and a technological point of view. The bibliography is extensive and well structured. Minor clarifications: Note 2: Some inaccuracies will need to be fixed. Eliminate the repetition of "gradually" in lines 94 and 96.”
The indicated text corrections have been made.
2. “Note 5: compared to what is said in the text, this definition seems too simplistic.”
The definition given in note 5 refers to the urban survey in the strict sense, understood as an operational tool used within a broader documentation process.
3. “Lines 128-136: It's a citation. Quote the original text and not the translation of a translation. You can find it on page 205 of the original text (https://archive.org/details/citiesinevolutio00gedduoft/page/204/mode/2up?ref=ol&view=theater&q=art+urban+planning). Note 12: Replace "vefore" with "before".”
The indicated text corrections have been made.
4. “Some sentences are a little too long and complex, reflecting a typically Italian syntactic construction. I would suggest simplifying them. Likewise, some terms do not seem perfectly adequate. A quick revision of the text is recommended to correct such inaccuracies.”
The syntactic construction of sentences that were too long and complex was simplified and a general revision of the text was carried out.
Round 2
Reviewer 3 Report
Comments and Suggestions for Authors
After a thorough review of the revised version of the article "Documenting Urban Morphology. From 2D representations to metaverse" by Alessandro Merlo and Gaia Lavoratti, Here are the key findings:
-
Definition and Use of Digital Twins in Urban Planning: The authors present a comprehensive understanding of digital twins, aligning well with the current applications in urban planning. They emphasize not just the visual representation but also the dynamic relationships and interactions within urban environments, which is consistent with contemporary views in the field.
-
Methods of Urban Space Modeling: The article accurately reflects current methods and technologies in urban space modeling. It discusses the evolution from 2D to 3D modeling and the integration of various data sources, aligning with modern urban planning practices.
-
Impact of New Technologies on Urban Space Perception and Planning: The authors' perspective on the impact of information technology, AI, and the metaverse on urban planning and perception is forward-thinking and aligns with current trends in the field. They discuss the potential of these technologies to transform urban analysis and planning, which is a view shared by many experts.
-
Potential Concerns and Ethical Considerations: The article also touches upon the ethical and social implications of using advanced technologies in urban planning, such as the potential detachment from physical spaces due to the rise of virtual environments. This perspective is important and relevant, reflecting a holistic understanding of the field.
In summary, the article by Alessandro Merlo and Gaia Lavoratti aligns well with the current knowledge and trends in urban planning and new technologies. It provides a comprehensive view of the evolution of urban space documentation, from traditional methods to advanced digital approaches, including the use of digital twins and the emerging concept of the metaverse.
I can point out potential weaknesses in the revised article. Here are some potential areas for critique:
Scope and Depth of Analysis: While the article provides a broad overview of technologies, it may lack more detailed analyses of specific case studies or examples of application. Presenting specific case studies or a deeper analysis of data could strengthen the author's arguments.
-
Future Predictions and Speculations: The article might overly rely on speculations regarding the future development and application of technologies, which can lead to overestimating their short-term impact.
It's important to remember that every academic article balances between theory and practice, and its value often lies in its ability to inspire further discussion and research.
The evaluated article fulfils this condition - it inspires scientific discussion. As it stands, recommends the article for inclusion in Land.
Author Response
Dear reviewer,
We thank you for your appreciation of our work and the further information provided. Regarding the case studies, we have implemented the description of the examples illustrated in the images by expanding the captions and inserting a further case study (fig. 7).
Best regards
